# Sucrosomial^®^ Iron: An Updated Review of Its Clinical Efficacy for the Treatment of Iron Deficiency

**DOI:** 10.3390/ph16060847

**Published:** 2023-06-06

**Authors:** Susana Gómez-Ramírez, Elisa Brilli, Germano Tarantino, Domenico Girelli, Manuel Muñoz

**Affiliations:** 1Department of Internal Medicine, University Hospital “Virgen de la Victoria”, 29010 Málaga, Spain; susanagram@yahoo.es; 2Scientific Department, Alesco S.r.l., Via delle Lenze, 56122 Pisa, Italy; e.brilli@pharmanutra.it; 3Scientific Department, Pharmanutra S.p.A., Via delle Lenze, 56122 Pisa, Italy; g.tarantino@pharmanutra.it; 4Department of Medicine, University of Verona, 37129 Verona, Italy; domenico.girelli@univr.it; 5Perioperative Transfusion Medicine, Department of Surgical Specialties, Biochemistry and Immunology, School of Medicine, Campus de Teatinos, 29071 Málaga, Spain

**Keywords:** iron deficiency, anemia, patient blood management, oral iron salts, intravenous iron, Sucrosomial^®^ iron, bioavailability, efficacy, tolerability

## Abstract

Iron deficiency (ID) and iron deficiency anemia (IDA) are highly prevalent worldwide. Oral iron salts, especially ferrous sulfate, are commonly used for the treatment of iron deficiency (ID). However, its use is associated with gastrointestinal side effects, thus compromising treatment compliance. Intravenous iron administration is a more costly and logistically complex alternative and is not risk-free, as infusion and hypersensitivity reactions may occur. Sucrosomial^®^ iron is an oral formulation consisting of ferric pyrophosphate conveyed by a phospholipid and sucrester matrix (sucrosome^®^). Intestinal Sucrosomial^®^ iron absorption is mediated by enterocytes and M cells, through the paracellular and transcellular routes, and occurs mostly as intact particles. These pharmacokinetic properties of Sucrosomial^®^ iron result in higher iron intestinal absorption and excellent gastrointestinal tolerance compared to oral iron salts. The evidence derived from clinical studies supports the use of Sucrosomial^®^ iron as a valid first option for the treatment of ID and IDA, especially for subjects who are intolerant or refractory to conventional iron salts. Newer evidence also demonstrates the effectiveness of Sucrosomial^®^ iron, with a lower cost and fewer side effects, in certain conditions usually treated with IV iron in current clinical practice.

## 1. Introduction

In 2019, an analysis of data from 204 countries revealed that there were ≈1.8 billion cases of anemia worldwide, with an age-standardized point prevalence of ≈23% [1,2]. In addition, iron deficiency anemia (IDA) was one important contributor to years lived with disability (YLD), with an age-standardized rate of ≈670 per 100,000 population [1,2] and with women aged 10–84 y showing higher YLD [1]. Globally, iron deficiency (ID) accounts for most anemia cases (males: 66%, females: 57%) [1]. As depicted in Table 1, limited supply or absorption, increased requirements, and/or increased iron losses are the most common causes of ID [1,2,3]. Thus, despite the significant reduction in the burden of anemia in recent years, the prevention and treatment of ID/IDA remain a global challenge [2,4]. 

Nevertheless, the prevalence of ID/IDA across the clinical setting is higher than among the general population, with great variability according to the underlying pathology [5,6,7,8,9,10,11,12,13] (Figure 1). Challenges in the treatment of ID/IDA include performing a correct diagnosis, finding and addressing the underlying cause, and selecting an iron product that safely meets the patient’s needs [14,15,16,17].
pharmaceuticals-16-00847-t001_Table 1Table 1Main causes of iron deficiency (adapted from Ref. [18], with permission).Increased iron demands:
  ○  Body growth (infancy and childhood)

  ○  Pregnancy and lactation

  ○  Recovery from blood loss (e.g., trauma, surgery, gastrointestinal bleeding)

  ○  Administration of erythropoiesis-stimulating agents (ESAs)
Limited iron supply or reduced absorption:
  ○  Malnutrition

  ○  Inappropriate diet:

    ▪  Deficiency of bioavailable iron and/or ascorbic acid

    ▪  Excess of dietary fiber, phenolic compounds from tea or coffee, and soya prod

  ○  Malabsorption syndromes:

    ▪  Autoimmune atrophic gastritis

    ▪  Gastric resection

    ▪  Bariatric surgery

    ▪  Inflammatory bowel disease

    ▪  Celiac disease and non-celiac gluten sensitivity

    ▪  *Helicobater pylori* infection

  ○  Medications:

    ▪  Histamine H2 receptor antagonists, proton pump inhibitors, antacids

    ▪  Antibiotics: tetracycline, penicillin, ciprofloxacin

    ▪  Anticonvulsants

    ▪  Cholestiramine

  ○  Increased hepcidin levels:

    ▪  Iron-refractory iron deficiency anemia (IRIDA)

    ▪  Amenia of chronic inflammation (ACI)

  ○  Deficiency of iron transport proteins:

    ▪  Heme oxygenase

    ▪  Divalent metal transporter 1 (DMT1)
Increased iron losses:
  ○  Major surgery and bleeding trauma

  ○  Gastrointestinal bleeding

    ▪  Peptic ulceration

    ▪  Neoplasia

    ▪  Inflammatory bowel disease

    ▪  Vascular malformations (e.g., angiodysplasia)

  ○  Genitourinary bleeding

  ○  Heavy menses and multi-parity

  ○  Multiple diagnostic phlebotomies (medical “vampirism”)

  ○  Blood donation

  ○  Dialysis (particularly hemodialysis)

  ○  Medications:

    ▪  Anti-inflammatory agents

    ▪  Platelet anti-aggregant agents

    ▪  Anticoagulant agents



We will review current definitions, approaches to diagnosis, and the therapeutic management of anemia and ID, with a special focus on updating the available evidence on the efficacy of Sucrosomial^®^ iron (SI) for the treatment of ID/IDA in different clinical settings [18].

## 2. Definitions of Anemia

An individual presents with anemia when their blood oxygen-carrying capacity is decreased due to a reduction in the number of circulating red blood cells (RBC), in the RBC hemoglobin (Hb) content, or both. This results in symptoms such as fatigue, weakness, dizziness, and shortness of breath, among others. The World Health Organization (WHO) has established hemoglobin (Hb) thresholds for the definition of anemia (Hb < 12 g/dL for women and Hb < 13 g/dL for men), as well as for its severity [19]. Though widely used, these anemia definitions may not be appropriate for some patient populations. In cardiac or major abdominal surgery, women presenting with “borderline” Hb concentrations (12.0 g/dL–12.9 g/dL) received more RBC transfusions (RBCT) and/or had a longer post-operative hospital stay compared to those with an Hb concentration of at least 13 g/dL [20,21]. Therefore, patients of either sex who are scheduled for a major surgical procedure and present with Hb < 13 g/dL should be considered as “anemic” and deserve pre-operative Hb optimization [22,23].

## 3. Iron Deficiency: Definitions and Diagnosis

Iron deficiency may be present even in the absence of anemia. In fact, the EMPIRE study (*n* = 7980) showed that ID (defined as ferritin < 30 ng/mL) was more prevalent than anemia in the adult Portuguese population (32% vs. 20%, respectively), irrespective of gender or age group [24]. According to WHO, “ID could be well advanced and causes clinical symptoms before Hb reaches the threshold for anemia” [19]. Therefore, ID should be considered as a disease in its own right and deserves appropriate diagnosis and treatment in populations at risk (e.g., individuals presenting with chronic fatigue) [17,25]. However, this is unusual as clinicians are not likely to suspect isolated ID and the diagnosis of ID is not always straightforward [26].

In the absence of anemia, a low mean corpuscular Hb (MCH; normal range 28–35 pg) or an increased red cell distribution width (RDW, normal range 11–15) is an indication of possible ID and the requirement for a ferritin test [4,22]. If there is no inflammation, a serum ferritin concentration <30 ng/mL provides an accurate definition of true ID, with high sensitivity (>90%) and specificity (>95%) [27] (Table 2). Should the patient present with inflammation (elevated C-reactive protein (CRP)) or impaired renal function (reduced glomerular filtration rate (eGFR) or elevated serum creatinine), the combination of serum ferritin < 100 ng/mL and low transferrin saturation (TSAT < 20%) also indicates the presence of ID (Table 2).

The finding of high serum ferritin (>100 ng/mL) and low TSAT (<20%) is usually catalogued as iron sequestration (also referred to as functional iron deficiency, FID) (Table 2). However, as an acute-phase protein, an elevated serum ferritin concentration in patients with an inflammatory status does not exclude ID. In these situations, some additional tests (such as the reticulocyte Hb content, the percentage of hypochromic RBC, the red cell size factor, the ferritin index, or the serum hepcidin concentration) may help in establishing an accurate diagnosis of ID, though they are not universally available [4,23,28,29,30].

Additionally, the administration of erythropoiesis-stimulating agents (ESAs) poses demand for iron by the bone marrow, which may surpass the iron mobilization capacity from the macrophages and results in FID (Table 2) [4,22,31]. Thus, most patients treated with erythropoiesis-stimulating agents (ESAs) will likely benefit from adjuvant iron supplementation.

Finally, individuals with a ferritin concentration within the normal range may still have low iron stores that are insufficient to meet their foreseeable iron needs. This could be the case for pregnant women or blood donors (ferritin < 50 ng/mL) or those scheduled for surgical procedures with expected moderate–high blood loss (ferritin < 100 ng/mL) (Table 2) [22].

## 4. Treatment Options for Iron Deficiency/Iron Deficiency Anemia

Should a patient present ID/IDA, it is mandatory to search for and address its cause, in addition to the initiation of the appropriate treatment with oral iron, intravenous (IV) iron, and/or blood transfusion, depending on the patient’s Hb levels, anemia tolerance, and co-morbidities. Whether it is a case of new-onset, recurrent, explained, or unexplained ID/IDA should also be considered at the time of selecting a therapeutic option.

### 4.1. Oral Iron

Oral iron salts, especially ferrous sulfate (FS, 100–200 mg elemental iron, 1–3 times a day), are commonly used to treat ID/IDA. However, the bioavailability of iron salts is low, especially for ferric formulations. Absorption is further impaired when administered at a high dose (by upregulation of hepcidin levels, which remain elevated over 24 h and tend to reduce the absorption of the next oral iron dose) [32] or when co-administered with food or drugs (e.g., antacids, proton pump inhibitors) [4]. As a result, its use is associated with gastrointestinal side effects, due to the interaction of unabsorbed iron with the enterocytes [33,34]. All these may undermine the compliance with treatment and its efficacy.

Nevertheless, provided that there is adequate tolerance, oral iron salts are recommended to treat uncomplicated, mild to moderate iron deficiency anemia. They should be given once daily at a low dose (40–50 mg elemental iron), to maximize efficacy without compromising safety. A single moderate iron dose (80–100 mg elemental iron) on alternate days is increasingly being considered as a feasible option, since it minimizes the “mucosal block” of absorption due to a transient hepcidin increase [22,35]. Though available evidence is restricted to young women [34,36], such a novel schedule for oral iron supplementation in ID treatment is being broadly accepted and is also included in recent guidelines [37].

As for surgical patients, those on the waiting list (“pre-operative” 4–8 weeks prior to surgery) might also benefit from this approach [38,39]. In contrast, due to surgery-associated inflammatory responses, the efficacy of post-operative oral iron administration is poor and it is therefore not recommended [22,35,40].

There are newer oral iron formulations with increased tolerance and improved absorption. One of these is ferric maltol, which has been shown to be safe and efficacious for the treatment of ID/IDA patients with inflammatory bowel disease or chronic kidney disease [41]. Data on its use in other clinical settings are lacking.

Sucrosomial^®^ iron is a nanoparticulate oral iron formulation that has been proven to be more tolerable and efficacious than oral iron salts for the management of ID/IDA in different clinical settings, even in the presence of inflammation [18]. In this updated review, we analyze newer evidence also demonstrating the effectiveness of SI, with lower costs and fewer side effects, in patients usually receiving IV iron.

### 4.2. Intravenous Iron

Intravenous iron represents a safe and effective alternative to oral iron for the treatment of ID or IDA, and its use is indicated for patients presenting with an intolerance, contraindication, or refractoriness to oral iron; inflammation; moderate-to-severe anemia; ongoing blood loss; and/or the use of erythropoiesis-stimulating agents. Among the currently available IV iron formulations, those allowing the administration of large single doses (e.g., ferric carboxymaltose (FCM) or ferric derisomaltose (FDM)) facilitate treatment and are preferred [22,40,42,43,44].

As for surgical patients, IV iron administration is considered (1) in those undergoing major procedures with predicted moderate-to-high perioperative blood loss; (2) when the timeframe for pre-operative Hb optimization ≤4 weeks; (3) in those presenting with ongoing blood loss; (4) as a co-adjuvant treatment with erythropoiesis-stimulating agents (ESAs); (5) for post-operative ID/IDA management [22,23,25]. Intravenous iron has been shown to be superior to oral iron, a placebo, or no treatment with a rising pre-operative Hb [45,46,47,48,49]. However, the effects of IV iron supplementation on blood transfusion are not as consistent, probably due to differences in study populations and treatment schedules [45,46,47,48,49]. This clearly indicates the need for further large-scale, methodologically robust, randomized controlled trials.

Compared to oral iron supplementation, intravenous iron acquisition and administration is a more costly and logistically complex alternative. It is not risk-free, thus requiring infusion and post-infusion monitoring, as infusion and hypersensitivity reactions may occur [50]. In accordance with the recommendations of the European Medicines Agency, “IV iron products should be administered only when staff is trained to evaluate and manage anaphylactic reactions, and resuscitation facilities are immediately available” [51]. Guidelines for the prevention and management of acute reactions to IV iron are available [52,53]. In addition, except for patients with chronic renal disease or chronic heart failure, the long-term safety of IV iron supplementation has not been conclusively established [7,54,55]. A more detailed analysis of the safety of the different options for iron replacement therapy is beyond the scope of this review and can be found elsewhere [17,50].

It is also important to consider that the timing of administration, severity of anemia, and underlying pathology influence the response to IV iron. In a study of 404 IDA patients, the maximal response to the administration of 500–1000 mg of FCM IV was observed 3–4 weeks post-dosing [56]. In elderly ID or IDA patients (≥65 years) undergoing major surgery (*n* = 251), the response to pre-operative IV iron increased with the severity of anemia and time of supplementation [57]. This seems to confirm the results from a previous study of 84 surgical IDA patients, where the response to the pre-operative administration of ≈1000 mg iron sucrose (IS) IV over 4–6 weeks prior to surgery was markedly influenced by the severity of anemia and the underlying pathology (inflammation, blood loss). The response was stronger in hysterectomy (more severe anemia, lesser inflammation) than in arthroplasty (mild to moderate anemia ± inflammation) or colorectal cancer (inflammation plus blood loss) patients [58].

### 4.3. Red Blood Cell Transfusion

Red blood cell transfusion (RBCT) could be indicated for rapidly increasing Hb concentrations in patients developing IDA, especially in those presenting with alarming symptoms (e.g., hemodynamic instability) and/or risk criteria (e.g., coronary heart disease). However, it must be borne in mind that it is a limited resource, of transitory efficacy, and not devoid of important associated complications.

A restrictive strategy for RBCT (Hb thresholds < 7–8 g/dL and/or presence of signs or symptoms of acute anemia) has been shown to be at least as effective as a liberal strategy (Hb < 9–10 g/dL) in most of the hospitalized patient populations, and it is strongly recommended by most guidelines [22,40,59,60,61].

For most hemodynamically stable IDA patients who may need transfusion, the administration of a single RBC unit may be a valid option. Thus, the transfusion of each RBC unit should be an independent clinical decision, and it requires the re-evaluation of the patient before prescribing the next one [22,40]. After achieving a safe Hb concentration by means of RBCT, additional iron supplementation should be considered [40].

## 5. Sucrosomial^®^ Iron for the Management of Iron Deficiency/Iron Deficiency Anemia in Different Clinical Settings

Sucrosomial^®^ Iron (SI), developed by Alesco srl (Pisa, Italy), represents an innovative oral iron-containing carrier, in which ferric pyrophosphate is enclosed in a phospholipid and sucrester matrix. Further stability and coating are obtained by adding other ingredients (tricalcium phosphate, starch), forming the “sucrosome” (Figure 2). This structure allows SI to be gastro-resistant and to avoid the interaction between iron and the intestinal mucosa, thus minimizing gastrointestinal side effects (Figure 2). To date, in vitro studies have shown that SI absorption can occur mainly through an hepcidin-independent pathway, as it is mostly taken up as a vesicle-like structure by enterocytes and M cells via the paracellular and transcellular routes, which are not restricted to the duodenum and proximal jejunum [18].

As previously reviewed [18], “SI has unique structural, physicochemical and pharmacokinetic characteristics, together with a high iron absorption rate and excellent gastrointestinal tolerability, compared to conventional oral iron salts. These properties make SI as a suitable formulation for the oral treatment of ID, even in clinical settings. Due to its behavior in the gastrointestinal tract, SI is well tolerated and highly bioavailable, compared to conventional iron salts, even in clinical settings where IV iron represent the usual therapeutic option”. These include ID/IDA patients with chronic kidney disease, inflammatory bowel disease, celiac disease, and cancer, and those undergoing bariatric surgery, among others [62,63,64,65,66,67]. Next, we review the efficacy and safety of oral SI for the management of ID/IDA in a variety of clinical settings.

### 5.1. Obstetrics and Gynecology

Iron deficiency and IDA are highly prevalent among pregnant women worldwide [1,2] and are associated with an increased risk of a poor pregnancy outcome (e.g., low birth weight, prematurity), low neonatal iron stores, preeclampsia, or post-partum hemorrhage [5], and even with an increased risk of peri-partum mortality when severe [68]. Reduced cognitive abilities, decreased physical performance, and impaired lactation are common features of postpartum IDA [5].

A consensus statement from the Network for the Advancement of Patient Blood Management, Thrombosis, and Hemostasis (NATA) recommends the routine antenatal administration of oral iron (30–60 mg/day) and folic (400 µg/day) to prevent the development of maternal ID and IDA, as well as the risk of newborn low birth weight [5]. This is in accordance with guidelines issued by WHO [69]. However, the gastrointestinal side effects of oral iron salts often compromise the adherence to treatment among pregnant women. Multivitamin and mineral compounds are better tolerated but most of them do not supply sufficient amounts of iron or vitamins B12, C, or D, especially for those already presenting with ID or IDA. In fact, the EMPIRE study revealed that most Portuguese pregnant women receive daily multivitamin and mineral products, as a source of iron supplementation, and 54% of them presented with ID or IDA symptoms [24].

In a prospective study, non-anemic pregnant women (Hb > 10.5 g/dL) were randomly assigned to treatments with no iron (control group; *n* = 20), FS 30 mg/day (FS group; *n* = 20), SI 14 mg/day (SI14 group; *n* = 20), or SI 28 mg/day (SI28 group; *n* = 20) from gestational week 12–14 up to postpartum week 6 (Table 3). At gestational week 28 and postpartum week 6, women from the SI28 group showed the highest Hb and ferritin concentrations (*p* < 0.01) and the lowest prevalence of IDA (10%). Moreover, there were no differences in these parameters among those receiving SI 14 mg/day and FS 30 mg/day, despite the lower iron dose. No women complained of gastrointestinal side effects. These data suggest the higher bioavailability of SI compared to FS [70] (Table 3).

More recently, the administration of 30 mg/day of Sucrosomial^®^ iron plus folic acid, vitamin D, vitamin B6, and vitamin B12 (Sideral^®^ Folico, Pharmanutra, Pisa, Italy), from week 6–8 to the end of pregnancy, was evaluated in 61 women. Hemoglobin increased significantly from 11.8 ± 0.7 g/dL at baseline to 12.2 ± 1.2 g/dL at delivery (*p* = 0.02), and 51 patients (84%) terminated the pregnancy with Hb > 10.5 g/dL. Treatment compliance was excellent, no adverse events were recorded, and all women gave birth at term [76].

In Spain, there is an ongoing, single-center randomized control trial (RCT) aimed at comparing the tolerability of oral SI (30 mg elemental iron/day) versus oral FS (80 mg elemental iron/day) in 180 women presenting with IDA during the second trimester of pregnancy (EudraCT Number: 2017-000994-35). In the interim analysis *(n* = 45), SI showed greater tolerability than FS, with a lower incidence of gastrointestinal adverse effects and, therefore, greater treatment compliance [77].

As for postpartum IDA (PPIDA), a multicenter prospective study included 60 women with mild (Hb = 9–11 g/dL) or moderate (Hb = 7–9 g/dL) PPIDA. All anemic women received oral SI for 60 days: 30 mg elemental iron/day for those with mild anemia and 60 mg elemental iron/day (10 days) plus 30 mg elemental iron/day (50 days) for those with moderate anemia. There was a substantial rise in Hb at day 60 (+3.6 g/dL; *p* < 0.01), which was already observed at day 10 (+2.1 g/dL, *p* < 0.01). Most patients (81%) experienced the correction of anemia (Hb ≥ 12 g/dL) at day 60. Similarly, a significant increment was observed in the percentage of patients with ferritin ≥30 n/mL (*p* < 0.05) or transferrin saturation ≥20% (*p* < 0.01) during the study period. No patient discontinued treatment due to gastrointestinal adverse events [78].

In an ongoing, prospective observational study in women undergoing transvaginal radiofrequency ablation of uterine fibroids, six out of ten women included, so far, have presented with anemia (mean Hb 10.3 g/dL) and received perioperative oral SI (30 mg elemental iron/day) up to at least 30 days after the procedure. Within one month post-intervention, the mean Hb increment was 2 g/dL, and anemia was corrected in 5 out 6 patients (mean Hb 12.6 g/dL). In the remaining one, her Hb increased from 8.6 to 11.0 g/dL [79].

### 5.2. Oncology

Onco-hematological patients are frequently affected by both ID and IDA [80]. Red blood cell transfusion, ESAs, and/or iron supplementation are generally prescribed for the treatment of IDA and chemotherapy-induced anemia (CIA). However, the safety and cost-effectiveness of the different iron compounds and their administration routes are still a matter of debate.

The presence of IDA and/or ID should be investigated in all cancer patients, especially in those undergoing cytotoxic chemotherapy, radiotherapy, or surgery. The assessment of ID/IDA should be performed both before and during treatment so as to assign patients to the most appropriate iron supplementation strategy [31].

The European Society of Medical Oncology (ESMO) guidelines recommend iron supplementation in patients with CIA (Hb ≤ 11 g/dL or Hb decrease of ≥2 g/dL from a baseline level of ≤12 g/dL) and absolute ID (serum ferritin < 100 ng/mL) [6]. For those with FID (normal or high ferritin and low TSAT), iron replacement therapy should be administered either as a monotherapy or as an adjuvant of ESA therapy [6].

According to the ESMO guidelines, oral iron should only be offered to patients with absolute ID and no inflammation (CRP < 5 mg/L). In contrast, patients with FID should be treated with IV iron, although its long-term safety in these patient populations has not been fully established [6]. Therefore, as its absorption does seem to be mostly hepcidin-independent, SI may represent an alternative for iron supplementation in anemic cancer patients with inflammation [31,62].

Oral SI administration (30–60 mg/day for 2–6 months) has been shown to be well tolerated and effective in increasing the Hb concentrations of anemic oncologic patients, with or without chemotherapy (Appendix A). In a retrospective study of patients presenting with moderate CIA (Hb 8–10 g/dL), with no ID or FID, and receiving ESA (Darbepoetin 500 μg/3 weeks), the hematological response was defined by an increment in Hb ≥ 2 g/dL and/or a final Hb ≥ 12 g/dL. No differences were observed in those receiving oral SI (30 mg/day, for 8 weeks; *n* = 33; response rate 70%) or IV ferric gluconate (125 mg/week, for 8 weeks; *n* = 31; response rate 71%) [66] (Appendix A). Additionally, no differences were observed in RBCT requirements (one patient in each group) or quality of life assessments. One patient in the SI group complained of gastrointestinal adverse effects, whereas two patients in the FG group experienced IV iron infusion reactions (Appendix A) (Table 3).

More recently, in a multicenter, open-label, phase III clinical trial, 60 cancer patients receiving ESA were randomly assigned (1:1) to receive adjuvant oral SI (30 mg elemental iron/day, for 12 weeks; SI group) or no iron supplementation (control group). Response to treatment was defined as an increase in Hb > 2 g/dL from baseline, without RBCT in the previous 28 days, or reaching an Hb ≥ 12 g/dL at week 12. Sucrosomial^®^ iron was well tolerated and there were more responders in the SI group compared to the control group (52% vs. 31%, respectively) (Appendix A). No differences in RBCT requirements were observed [81].

### 5.3. Nephrology

Iron deficiency is one of the main causes of anemia in patients with chronic kidney disease (CKD), and iron supplements, along with ESAs, constitute the basis of its treatment, both for CKD patients not on dialysis (ND-CKD) and those who are hemodialysis-dependent (HD-CKD). However, disparities exist in the guidelines and position papers regarding treatment modalities for CKD-related anemia across the world [7,82].

Overall, there are no differences in mortality and adverse effect rates related to the iron supplementation route in CKD patients. However, in a meta-analysis of 24 studies, including 3187 patients, hypotensive reactions were more frequent among those receiving IV iron, whereas gastrointestinal adverse events were more frequent among those receiving oral iron salts [83].

An RCT in 128 anemic CKD patients suggested a possible higher incidence of cardiovascular events and hospitalization for infection in the IV iron sucrose arm compared to the oral FS arm [84]. In contrast, no differences in the infection or cardiac event rates between groups were registered (3.9%, 3.3%, and 3.8%, respectively) in an RCT of 626 anemic ND-CKD patients, with ID and not receiving ESA therapy, that compared iron supplementation with high-dose IV FCM (500–1000 mg/4 weeks), low-dose IV FCM (200 mg/4 weeks), and oral FS (200 mg/day) for 56 weeks [85]. An Hb increase of at least 1 g/dL was achieved by 21.6% of patients from the oral iron group, suggesting the benefit of the early consideration of IV iron supplementation in this clinical setting [86]. In a very large series of HD-CKD patients (*n* = 58,058), IV iron doses >400 mg/month were associated with higher death rates due to cardiovascular events [87]. However, this was not confirmed by a pre-specified secondary analysis of the PIVOTAL trial, which found no significant difference between high- and low-dose IV iron supplementation regarding the incidence of infection (46.5% versus 45.5%, respectively) or hospitalization for infection (29.6% versus 29.3%, respectively) [88].

In anemic ND-CKD patients, non-comparative studies have shown oral SI supplementation (30 mg/day), with or without ESAs, to be efficacious for the maintenance and/or improvement of Hb, ferritin, and TSAT concentrations through 3- to 18-month follow-up periods, whereas the incidence of gastrointestinal side effects was almost negligible (Appendix A). In an open-label RCT, 99 ND-CKD patients with IDA (Hb ≤ 12 g/dL, ferritin ≤ 100 ng/mL, TSAT ≤ 25%) were assigned (2:1) to receive oral SI (30 mg/day) for 3 months or IV FG (125 mg/week; cumulative dose, 1000 mg) [65]. The follow-up period was 4 months. After 3 months of treatment, there were no differences in Hb concentrations between groups (11.4 g/dL vs. 11.7 g/dL, respectively). Compared to the oral SI group, higher ferritin concentrations (86 ng/mL vs. 239 ng/mL, respectively; *p* < 0.05) and a higher incidence of adverse events (*p* < 0.001) were recorded in the IV FG group (Table 3). Therefore, short-term, low-dose oral SI seems to be as efficacious as IV FG for the correction of anemia in ND-CKD patients. In addition, the observed ferritin concentrations after a 3-month course of supplementation with SI suggest that iron overload during its long-term use is unlikely. This seems to be confirmed by two small cases series (*n* = 67) of ND-CKD with 3–5 patients, where the administration of SI (30 mg elemental iron/day) for 12–18 months resulted in maintained or increased Hb levels, without major increments in ferritin at the end of the study period (final mean ferritin 99–116 ng/mL) (Appendix A) [89,90]. Preliminary data also suggest the similar efficacy of oral SI and IV FG, with or without ESA, for the treatment of anemia in HD-CKD patients *(n* = 58) (Appendix A).

Regarding the costs of iron supplementation, a 4-year budget impact model (2017–2020) has been applied in three CKD patient populations (pre-dialysis, peritoneal dialysis, and post-transplant) refractory to oral iron salts [91]. Using this model, it was estimated that replacing 10% of IV iron administrations (either IS or FCM) with oral SI would result in over EUR 750,000 in savings for the Spanish healthcare system [91]. Additionally, a cost minimization analysis of oral SI, compared with IV FG, for the treatment of anemia in ND-CKD patients from an Italian societal perspective showed that oral SI could result in significant savings (EUR 1191 per patient/cycle) and allowed the identification of some implications for future research [92].

### 5.4. Gastroenterology

#### 5.4.1. Inflammatory Bowel Disease

As depicted in Figure 1, ID and FID may affect 36–75% of IBD patients [18], and both IDA and ACD are also highly prevalent among them [93]. However, the absorption of iron salts is poor in IBD patients, due to the inflammation-induced increase in circulating hepcidin concentrations, and the unabsorbed iron may lead to oxidative stress (due to Fenton reactions) and to microbiota alterations, both resulting in worsening disease symptoms (flares) [93].

In both experimental animal models and patients, the incidence and severity of these detrimental effects depended on the formulation used [93,94,95]. In animal models, mice fed for two weeks with diets containing SI, as the sole iron source, showed a decrease in the abundance of the *phylum Proteobacteria*, which contains many pathogenic species, and an increase in short-chain-fatty-acid-producing bacteria, such as *Lachnospiraceae*, *Oscillibacter*, and *Faecalibaculum*. None of these changes were observed when dietary SI was replaced with FS [96].

In IBD patients, microbiota alterations and gastrointestinal side effects were less pronounced with IV iron sucrose compared to oral FS [93], and current guidelines recommend the preferential use of IV iron for the treatment of IBD-associated ID/IDA [9]. However, initial clinical data suggest that oral iron formulations with improved absorption and tolerability, such as SI or ferric maltol, may represent a safe and efficacious therapeutic option in these patient populations [97].

In two case series of IBD patients with mild-to-moderate IDA (*n* = 76, including 46 intolerant to ferrous sulfate), SI (30–60 mg/day for 2–3 months) was shown to be efficacious in increasing Hb concentrations (+0.6 g/dL), as well as ferritin and TSAT levels, with very few gastrointestinal side effects [98,99] (Appendix A). In a study comparing 3-month iron courses, SI (30–60 mg/day) resulted in higher Hb increments, compared to FS (105 mg/day) or no iron (mean Hb changes: +1.9 g/dL, +0.9 g/dL and +0.6 g/dL, respectively; *p* < 0.01), despite lower elemental iron doses (Appendix A) [100]. In patients with ulcerative colitis, the efficacy of oral SI (60 mg elemental iron/day for 60 days, plus 30 mg/day for 30 additional days) in increasing Hb concentrations was similar to that of IV FCM (1000 mg, at baseline) (mean Hb change: +1.1 g/dL vs. +1.5 g/dL, respectively; *p* = NS) [71] (Table 3).

From the Spanish healthcare system’s perspective, in a 4-year budget impact model (2018–2021) developed for Crohn’s disease patients with ID and intolerant to oral iron, a progressive increment in SI use (from 0% to 10% of market shares), with a concomitant reduction in that of IV iron, was found to lead to savings of EUR ≈ 250,000 over the study period [101].

#### 5.4.2. Celiac Disease

Celiac disease (CD) is an intestinal autoimmune pathology that presents with variable clinical manifestations, including gastrointestinal symptoms (loose stools, weight loss, dyspepsia, flatulence, among others), as well as osteopenia, fertility problems, and iron deficiency [102]. The latter may persist despite an adequate gluten-free diet [102].

The efficacy of SI supplementation (30 mg/day for 3 months) for the treatment of IDA in 24 CD patients, intolerant to oral FS, was prospectively evaluated (NCT02916654) [63]. Another 19 naïve CD patients with IDA received oral FS (105 mg/day) (*n* = 19). Both treatments led to significant improvements in iron parameters and to similar rates of anemia correction (70% vs. 82%, respectively; *p* = ns), despite the fact that the elemental iron dose with SI was one third of that with FS (Table 3). In addition, patients from the SI group reported less severe gastrointestinal adverse effects and a greater improvement in general well-being, compared to those receiving FS [63].

In a pilot study of non-CD, gluten-sensitive patients with IDA *(n* = 28), a 3-month course of oral SI (30 mg/day for 15 days, plus 15 mg/day for 75 days) was well tolerated and resulted in significant increments in both Hb (+2.8 g/dL) and ferritin concentrations (+11 ng/mL) [103] (Appendix A).

#### 5.4.3. Autoimmune Atrophic Gastritis

In autoimmune atrophic gastritis (AAG), autoantibodies against gastric parietal cells and/or the intrinsic factor led to iron and/or cobalamin deficiency, and subsequently to IDA and/or megaloblastic anemia [104]. In fact, AAG accounts for the inefficacy of oral iron treatment in ≈25% of IDA patients, who chose to be switched to IV iron supplementation [105].

The efficacy of an 8-week course of oral SI (120 mg/daily, either fasting or during meals) was prospectively assessed in 20 consecutive AAG women presenting with newly diagnosed, mild-to-moderate IDA (Hb 10.5 g/dL, ferritin 7 ng/mL, TSAT 8%) [72]. Three patients did not complete the iron course—two due to intolerance and one due to a lack of compliance. In the remaining 17 patients, oral SI supplementation significantly improved Hb (+2 g/dL), ferritin (+20 ng/mL), and TSAT (+10%), with respect to baseline (Table 3).

#### 5.4.4. Bariatric Surgery

Nutritional deficiencies, including ID, are common after bariatric surgery (BS), especially in those with a malabsorptive component. Iron deficiency develops as a consequence of limited gastric acid secretion and the exclusion of intestinal areas of iron absorption. Its prevalence increases over the years, and women of reproductive age are the group at highest risk [10]. Due to the underlying mechanism, iron salts are poorly tolerated and of limited efficacy in treating BS-associated ID, leading to most patients being switched to parenteral iron therapy.

The efficacy of oral SI, as an alternative treatment option for BS-associated ID, was evaluated in a case–control study of 40 women of reproductive age who were on maintenance treatment with IV iron sucrose supplementation (300 mg every 3 months). Women were divided into two groups: twenty women were switched to oral SI (28 mg/day for 3 months), and another 20 received the corresponding iron sucrose IV (300 mg). No between-group differences in Hb or iron parameters were observed either at baseline or after a 3-month follow-up [64] (Table 3). Thus, for patients with BS-associated ID and requiring parenteral iron supplementation, SI could be an alternative option for maintenance therapy.

The efficacy and tolerability of a Sucrosomial^®^ mineral and vitamin food supplement for special medical purposes (Sideral-Med^®^, Pharmanutra, Pisa, Italy), for the treatment of micronutrient deficiencies (iron, calcium, magnesium, zinc, water-soluble and fat-soluble vitamins) after one anastomosis gastric bypass (OAGB), are being prospectively evaluated [106]. Patients are included if they have undergone OAGB at least six months ago, are on standard multivitamin–multimineral supplementation, and present with one or more than one of the above-mentioned micronutrient deficiencies. Included patients receive a 6-month course of Sideral-Med^®^, twice daily. Overall, 17 patients have already entered the study. Preliminary results after 3-month supplementation show an increase in vitamin B_12_, vitamin D, and folate levels, and a slight improvement in ferritin, while maintaining Hb concentrations. Gastrointestinal symptoms (nausea, reflux, vomiting, diarrhea, or constipation) have been reported by 55% of patients, but not leading to treatment discontinuation.

### 5.5. Cardiology

Iron is not only needed for erythropoiesis, but also for oxygen utilization in energy production (respiratory chain) in skeletal and cardiac muscle cells. In congestive heart failure (CHF), ID/IDA affects nearly 50% of patients and is independently associated with poor clinical outcomes, affecting physical performance, quality of life, and the risk of mortality [8]. In this patient population, the detection and treatment of ID, with or without anemia, is recommended by guidelines issued by the European Society of Cardiology [107].

Although not completely understood, the etiology of ID in CHF patients seems to be multifactorial, including low-grade inflammation [4,14,55], a reduced iron supply (malnutrition) or absorption (intestinal edema), anti-thrombotic therapy-associated blood losses [108], and/or intestinal dysbiosis [63]. All these make oral iron therapy of little clinical value in treating ID in CHF, as demonstrated for oral iron polysaccharide (150 mg, bid) in the IRONOUT study [109], and current guidelines recommend the IV route for iron supplementation [107].

However, the possible role of oral SI supplementation was evaluated in 50 CHF patients with a reduced ejection fraction (CHFrEF) and ID iron deficiency, with or without anemia, prospectively assessed in a proof-of-concept study [73] (Table 3). Patients were matched regarding clinical parameters, exercise capacity, and quality of life, and assigned to receive SI (28 mg/day for 3 months; SI group, *n* = 25) or no iron (control group, *n* = 25). All patients were on optimal stable CHF therapy and followed for 6 months. At 3 and 6 months, SI was associated with significant increases in hemoglobin, serum iron, and serum ferritin concentrations, along with a significant improvement in the 6-min walking distance (6MWD) and Kansas City Cardiomyopathy Questionnaire (all *p* < 0.01), even after adjustment for baseline parameters. Ten patients (4 in the SI group and 6 in the control group) experienced worsening CHF (OR = 0.51, 95%CI, 0.41–6.79, *p* = 0.482). Only one patient from the SI group discontinued treatment due gastrointestinal adverse events (diarrhea).

Similar data were reported by two small case series (*n* = 19) (Appendix A), but further, appropriately sized studies are warranted. The comparative efficacy of oral SI, ferrous bisglycinate chelate, and IV FCM for the treatment of ID in CHF is being assessed in two ongoing randomized controlled trials (PREFER-HF study, NCT03833336; IVOFER-HF study, EudraCT Number: 2017-005053-37), although recruitment has been severely limited by the COVID-19 pandemic.

In an observational study of 50 anemic patients after percutaneous coronary intervention [110], who received a 3-month course of iron supplementation, SI (30 mg/day) (*n* = 25) was as effective as FS (105 mg/day) (*n* = 25) in increasing Hb concentrations, though with fewer gastrointestinal side effects (0% vs. 32%, respectively) (Appendix A).

Finally, ID is highly prevalent among patients presenting with pulmonary artery hypertension (PAH) and is associated with worse clinical conditions. In a small series of 31 patients with idiopathic PAH, 6 presented with IDA and 16 with ID. Oral SI supplementation (30 mg elemental iron/day) was prescribed to 5 out of 22 patients with ID. After 16 weeks of oral SI, there was a slight increase in RBC and ferritin, which was associated with a significant improvement in functional capacity (6-min walking distance) and a trend toward lower systolic pulmonary artery pressure, which was not seen in ID patients not receiving SI supplementation [111]. The potential of SI as a therapeutic tool for ID PAH patients merits further investigation.

### 5.6. Internal Medicine

Anemia is a frequent condition among hospitalized patients in the internal medicine (IM) ward and may compromise their clinical outcomes. However, the detrimental role of anemia in this patient population has not been investigated as frequently as in other hospitalized patients (e.g., surgical or critically ill patients). Data from 771 consecutive IM patients have shown an association between anemia on admission (67%) and an increased risk of in-hospital mortality (RR 1.82) [112]. There was also a high prevalence of ID (19%) and IDA (40%) among IM patients [112]. More recently, in a series of 681 patients admitted to an IM ward, anemia on discharge was shown to be a predictor of short-term (6 months) and long-term (1, 3, and 5 years) mortality, independently of the reason for admission [113]. Therefore, all institutions should implement a protocol for the timely diagnosis and appropriate treatment of anemia in patients admitted to the IM ward.

In three small series (*n* = 55) and three observational studies of patients with IDA of different origin (*n* = 137), including inflammatory diseases (systemic erythematosus lupus, rheumatic fibromyalgia, connectivitis), SI supplementation has been shown to ameliorate/correct anemia and improve iron stores, with efficacy that was superior to that of FS and similar to that of IV iron. Additionally, SI, but not FS, was associated with a significant reduction in the levels of inflammatory markers, such as CRP, and the erythrocyte sedimentation rate (ESR) (Appendix A).

Patients with myelodysplastic syndrome (MDS) frequently present with anemia of chronic disease, which is a subsidiary of IV iron supplementation [6]. However, in a retrospective series of 135 MDS patients with low-risk refractory anemia receiving subcutaneous epoetin-α (40,000 IU/session), the comparative efficacy of oral SI (28 mg/day), IV FG (62.5 mg, along with epoetin-α), or no iron in supporting the erythropoietic response to ESA was studied over 3 months [74]. The hematological response was defined by an Hb increment ≥ 1.5 g/dL at the end of the treatment course, and there was no difference in the response rate between the two iron-supplemented groups (50% and 43%, respectively), thus suggesting the efficacy of oral SI supplementation (Table 3).

Bleeding is a frequent etiology of anemia in the IM ward, and it is usually managed with RBCT, although alternative therapeutic strategies could be applied in patients without hemodynamic instability. Patients with gastrointestinal or gynecologic bleeding-induced IDA, without inflammation or malignancy but intolerant/refractory to FS, were included in a recent RCT [75]. They presented with moderate-to-severe IDA and were randomly assigned to receive oral SI (120 mg/day for one month; with or without food or antacid therapy; SI group; *n* = 45) or IV FG (62.5 mg/day until replenished total ID; FG group; *n* = 45). There were no between-group differences in either baseline (8.5 g/dL vs. 8.2 g/dL, for SI and FG groups, respectively; *p* = NS) or final Hb concentrations (12.0 g/dL vs. 12.5 g/dL, after 4 weeks, respectively; *p* = NS), and no patient received RBCT (Table 3). Similarly, there was no difference in the incidence of treatment-related adverse events (26% vs. 20%, respectively; *p* = NS) (Table 3). In addition, treatment with oral SI resulted in significant cost savings (−180 EUR/patient), compared to that with IV FG. Overall, results from this RCT seem to support those of several observational studies using oral SI for the treatment of IDA of different etiologies (mostly bleeding) (Appendix A).

In another multicenter RCT, 300 patients with moderate-to severe IDA (Hb < 11 g/dL, ferritin < 30 ng/mL), due to gastric (44%) or intestinal (56%) bleeding, were equally divided into six groups and received one of six different oral iron formulations (60 mg of elemental iron/day, for 12–24 weeks) [114]. From week 6 onwards, patients from the SI group showed significantly higher Hb increments, compared to those receiving any other oral iron compound (FS, iron saccharate, micronized ferric pyrophosphate, heminic bisglycinated iron, or bisglycinated iron) and irrespective of patients’ inflammatory status. At week 24, the Hb concentration in the SI group was 13.2 g/dL (12.5 g/dL for subgroup with inflammation at baseline). Therefore, SI emerged as the oral iron formulation with the fastest effect and greatest efficacy in correcting IDA, especially in patients with inflammation. Regarding tolerability, FS had the highest rate of gastrointestinal adverse events compared to any other formulation tested (30% vs. 6–12%; *p* < 0.05) [114].

### 5.7. Pediatrics

Microcytic anemia is the most common form of anemia in the pediatric population, affecting 17% of children under 5 years old in industrialized countries [115]. In a retrospective study, 25 children presenting with uncomplicated microcytic IDA and aged <16 years were treated with SI (mg elemental iron/day). Compared to baseline, after 2 months of SI supplementation, significant increments in Hb (8.9 g/dL vs. 10.4 g/dL, respectively; *p* < 0.01), MCV (62 fL vs. 71 fL, respectively; *p* < 0.01), and sideremia (18 mg/dL vs. 32 mg/dL, respectively; *p* < 0.05) were observed. There was also a non-significant trend toward a ferritin increment (5 ng/mL vs. 8 ng/mL, respectively; *p* = 0.48), thus suggesting that the absorbed iron was used primarily for Hb synthesis and highlighting the need for treatment continuation until Hb and ferritin levels are fully restored [116].

As in adults, anemia is the most common extra-intestinal manifestation of IBD in children. In a single-center, retrospective study, 76 out of 107 IBD children (71%) presented with anemia: 45 mild (Hb < 12 g/dL), 24 moderate (Hb < 10 g/dL), and 7 severe (Hb < 8 g/dL). At the physician’s discretion, those with mild-to-moderate anemia (*n* = 55) received oral SI (3 mg/kg/day), whereas those with moderate-to-severe (*n* = 21) anemia received IV FCM (15 mg/kg/dose) followed by oral SI supplementation. After the first 3 months of iron supplementation, 23 remained anemic (30%), and continued receiving oral SI. For the whole series mean, the average oral SI treatment was 8 ± 6 months. Thus, the administration of oral/parenteral iron replacement therapy successfully re-established Hb levels in most anemic pediatric patients. No adverse effects were recorded on oral SI therapy (confirming the excellent safety profile of oral SI) and only one patient developed an acute, moderate reaction during IV FCM administration [117].

Iron-refractory iron-related anemia (IRIDA) is a rare autosomal recessive disorder caused by the loss of function mutations on the *TMPRSS6* gene, which encodes Matriptase 2 (MT2), an upstream negative regulator of hepcidin [118]. The currently recommended treatment is parenteral iron supplementation, based on the fact that increased hepcidin levels typically associated with IRIDA prevent, by definition, adequate oral iron absorption. However, SI was shown to be effective in correcting IRIDA in one patient, who only partially responded to IV FG [67]. Such a rare disorder represents a useful model for the study of iron absorption when hepcidin levels are constitutively high and not due to inflammation. Indeed, Asperti and colleagues evaluated the hematological response to different oral iron preparations in the IRIDA “Mask” mouse model. Of note, only SI (but not FS) was shown to increase hemoglobin levels in msk/msk mice. Thus, SI may represent a promising option for oral iron supplementation in IRIDA patients, probably because of its alternative absorption pathway [119].

### 5.8. Patient Blood Management

Patient blood management (PBM) is a patient-centered, systematic, evidence-based approach that actively manages and preserves a patient’s own blood, thus improving safety and outcomes while promoting patient empowerment [120]. PBM relies on a number of strategies grouped into three pillars, which can be applied pre-, intra-, and post-operatively: (1) optimize erythropoiesis (including red cell mass and iron stores), (2) minimize blood loss (surgical and iatrogenic) and address coagulopathy, and (3) harness and optimize the patient’s tolerance to anemia while treatment is initiated, thus allowing restrictive RBCT criteria. According to WHO, there is an urgent need to implement PBM programs worldwide [121], and the improvement of perioperative erythropoiesis is a central pillar of PBM.

Overall, around one third of patients scheduled for major elective procedures presented with pre-operative anemia [13], which has been shown to be independently associated with poor clinical outcomes (increased risk of morbidity, mortality, and readmission; increased RBCT requirements; and longer length of hospital stay) and enhancing the deleterious effects of blood loss and RBCT [12,122]. Post-operative anemia may develop in up to 80–90% of surgical patients and negatively affects their functional recovery and quality of life [123]. Recommendations on the management of anemia in surgical patients have been recently updated [23].

Iron deficiency or FID accounts for the majority of cases of pre-operative anemia (up to 70%). In turn, post-operative anemia may develop or be aggravated by perioperative bleeding and the inflammatory response to surgery [13]. Isolated hematinic deficiencies are also frequent and may hinder the optimization of pre-operative Hb concentrations and/or recovery from post-operative anemia [13]. This is especially important for non-anemic iron deficiency (NAID), as patients presenting with NAID are at a higher risk of post-operative transfusion, morbidity, and mortality [124,125,126,127].

Therefore, as modifiable risk factors, both pre-operative anemia (Hb < 13 g/dL, irrespective of genders) and hematinic deficiencies should be appropriately managed prior to any major surgery [22,23]. However, the role of oral iron salts in treating ID, FID, or IDA in the pre-operative setting has not been fully established, and evidence derived from randomized controlled trials denies the utility of the post-operative administration of oral iron salts [35].

In a study of patients undergoing hip replacement surgery, the impact of pre-operative supplementation with oral SI on RBCT requirements, LHS, and post-operative anemia recovery was retrospectively evaluated [128]. Overall, 300 paired-matched, non-anemic patients (Hb > 12 g/dL in women or >13 g/dL in men) entered the study. Oral SI (30 mg elemental iron/day, for 21–29 days) was offered to 100 patients with ID (ferritin < 100 ng/mL) or FID (ferritin > 100 ng/mL plus CRP > 0.5 mg/dL or TSAT < 20%). Compared to 100 demographic- and laboratory-matched patients who had not received iron, SI supplementation led to lower RBCT requirements (0 units vs. 7 units, respectively), shorter lengths of hospital stay (4 days vs. 6.5 days, respectively; *p* < 0.01), and higher Hb concentrations one month after discharge (13.4 ± 1.5 vs. 10.2 ± 1.2, respectively; *p* < 0.01). Interestingly, there were no differences in post-operative data from the treated group when compared to those from 100 non-anemic, non-ID patients [128] (Table 4). In a recent RCT on patients scheduled for prosthetic hip or knee surgery, pre-operative supplementation with SI (30 mg/day for 30 days) improved Hb at admission and iron status in those ≥65 y as compared to no treatment [129].

Recently, the efficacy and safety of SI in increasing pre-operative Hb have been tested in a pragmatic, single-center, randomized study of elective cardiac surgery (CS) patients. One thousand consecutive patients with pre-operative Hb ≤ 14.5 g/dL were recruited. Patients were randomly assigned to receive a one-month course of Sucrosomial^®^ iron (60 mg/day, treatment group; *n* = 500) or no treatment (control group, *n* = 500) prior to elective heart surgery. Primary endpoints were the Hb concentration on the day of hospital admittance and blood transfusions (ClincalTrials.gov NCT03560687) [130]. Admission hemoglobin was higher by 0.67 g/dL in the treatment group (*p* < 0.001). Of note, the observed Hb increment with SI was similar to that previously observed with the pre-operative administration of IV ferric carboxymaltose (+0.84 g/dL) [131] or IV ferric derisomaltoside (+0.95 g/dL) [132] in anemic subjects scheduled for CS. The transfusion rate was lower in the treated group compared to the control group (35.4% vs. 64.6%, respectively; *p* < 0.001), as was the average number of transfused units (0.95 U/patient vs. 2.03 U/patient, respectively; *p* < 0.01) [130] (Table 4).

A post-hoc analysis of the abovementioned study, including 594 patients with known baseline Hb concentrations [133], showed that the improvements in these three parameters were more prominent in the subgroup with Hb < 13 g/dL receiving pre-operative oral SI, while there was no between-group difference in Hb drift. The tolerability of SI was excellent (98%), and the estimation of the overall cost-effectiveness of SI administration suggested savings of 156 EUR/patient with respect to the control group.

As for post-operative anemia, 106 consecutive patients presenting with IDA on post-operative day (POD) 10 after CS (T1) were treated with SI (*n* = 54) or FCM *n* = 52) [134]. Patients received a single dose of FCM (1000 mg) on POD 10 or oral SI (120 mg/day from POD10 to POD20 plus 30 mg/day from POD21 to POD30). The two groups were comparable in terms of age, left ventricular ejection fraction (LVEF), and laboratory parameters at baseline. Hemoglobin increased at POD20 and POD30, with no significant differences between groups. TSAT displayed similar behavior, showing a significant increase at POD30 (*p* < 0.001), although the increase was greater with FCM. Ferritin, elevated at baseline for inflammation due to CS, decreased for POD30 with SI, while it significantly increased in the FCM group. Over time, CRP levels decreased (*p* < 0.001), while 6MWT values improved (*p* < 0.001), without significant differences between groups (*p* = 0468). Brain natriuretic peptide concentrations were reduced during the follow-up, with both treatments, but not significantly [134] (Table 4).A similar improvement in post-operative Hb recovery was registered in a small RCT of 51 patients undergoing abdominal aortic aneurism repair comparing SI (30 mg/day for 30 days) versus no treatment (Hb increment: 1.9 g/dL vs. 0.4 g/dL, respectively) [135] (Table 4).
pharmaceuticals-16-00847-t004_Table 4Table 4Sucrosomial iron (SI) administration in surgical patients (4 studies, 1457 patients).Author(Year) [Ref]Study TypePatientsTreatmentCompound (Dose)DurationBasalHb(g/dL)FinalHb(g/dL)BasalFerritin(ng/mL)FinalFerritin(ng/mL)ABT Rate (%)LOS(Days)GISide Effects*Orthopedic surgery*Scardino et al.(2019) [128]ObsTHR ID ^a^THR IDTHR non-IDpreoperativeNo iron (*n* = 100)SI (30 mg/day, for 3–4 weeks; *n* = 100)No iron (*n* = 100)13.513.514.810.2 ^b^13.312.86665160---------7006.544---No---*Cardiovascular surgery*Pierelli et al.(2021) [130]RCT1000 patientspreoperativeSI (60 mg/day, 30 days; *n* = 500)Control (standard preparation; *n* = 500)------13.913.3------18416065 ^c^3513151.8%---Venturini et al.(2022) [134]Obs106 patients postoperativeSI (120 mg/day, 10 days + 30 mg/day, 10 days; *n* = 54)FCM (1000 mg IV, single dose; *n* = 52)10.110.112.012.5411386220689------------NoNoLucertiniet al.(2020) [135]RCT51 patientsAAARpostoperativeSI (30 mg/day, 30 days; starting PO10; *n* = 26)Control (no iron; *n* = 25)9.39.311.29.7------------00------NoNoAAAR, abdominal aortic aneurysm repair; FCM, ferric carboxymaltose; FS, ferrous sulphate; ID, iron deficiency; Obs, observational; THR, total hip replacement. ^a^ ID defined by ferritin < 100 ng/mL, or ferritin > 100 ng/mL but C-reactive protein > 3 mg/L and transferrin saturation (TSAT) < 20%. ^b^ Final Hb was measured at POD30; SI administration resulted in cost saving 1763 Є/pt, ID-SI vs. ID-no iron ^c^ Compared to control, SI treatment resulted in reduced transfusion rate (35% vs. 65%, respectively; *p* < 0.001) and transfusion index (0.95 units/pt vs. 2.03 units/pt, respectively; *p* > 0.001). 


Overall, data from these studies suggest a role for SI in PBM, as it was shown to be as effective as IV iron for the treatment of perioperative ID, while offering organizational and cost benefits for the healthcare system. This has been acknowledged in a recent Editorial regarding the guidance addressing perioperative anemia published by the Centre for Perioperative Care (CPOC) in the UK, where the authors stated that “Sucrosomial^®^ iron (Alesco S.r.l., Pisa, Italy) is an exciting development that could change the landscape of anaemia treatment…. Early evidence in cardiac surgery and non-surgical bleeding-induced anaemia suggests good results, but further evaluation is needed. If Sucrosomial iron becomes widely available and proves cost-effective, it could be given early in an anaemia treatment pathway by primary care, potentially reducing the burden of anaemia and requirements for intravenous iron in secondary care” [136] (p. 117).

## 6. Conclusions

The analysis of the available data seems to support oral SI as a valid therapeutic option for iron supplementation. Compared to oral iron salts, SI has been shown to be more convenient, efficacious (better Hb recovery and/or replenishment of iron stores at lower doses), and tolerable (fewer gastrointestinal adverse effects) in multiple clinical scenarios (Table 3 and Table 4). As depicted in Table 5, data from ≈2000 patients receiving oral SI supplementation, including a significant proportion of patients intolerant to FS, reveal a mean incidence of gastrointestinal adverse effects of 4.5%, which favorably compares with the much higher incidence reported for oral iron salts [33,34]. Newer evidence also demonstrates the effectiveness of SI, with lower costs and fewer side effects, in patients usually receiving IV iron (e.g., CKD, IBD, cancer, post-operative anemia, bariatric surgery, CHF) (Table 3 and Table 4).

Thus, the administration of oral SI emerges as a safe and efficacious first option for the treatment of uncomplicated ID, especially for patients who are intolerant and/or refractory to traditional oral iron formulations. Moreover, oral SI should also be considered as an alternative option for the initial and/or maintenance treatment of ID, in certain conditions usually treated with IV iron in current clinical practice, except in patients for whom immediate iron replenishment is required (e.g., IDA patients scheduled for colorectal cancer surgery or undergoing hip fracture repair). However, as SI is commercialized as a nutritional supplement, there may be reimbursement issues in some countries.

Nevertheless, to confirm the promising results of oral SI supplementation in different clinical settings, appropriately sized randomized control trials, with sufficient follow-up, are warranted.

## Figures and Tables

**Figure 1 pharmaceuticals-16-00847-f001:**
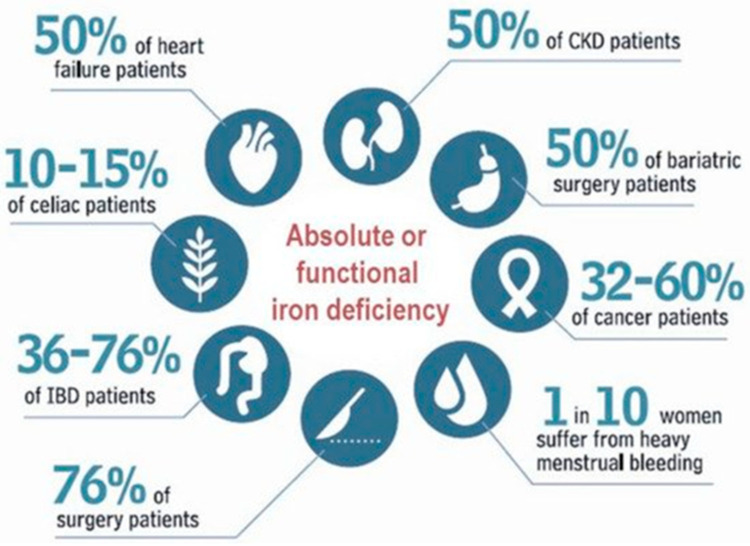
Estimated prevalence of iron deficiency in different clinical settings (taken from Ref. [18], with permission).

**Figure 2 pharmaceuticals-16-00847-f002:**
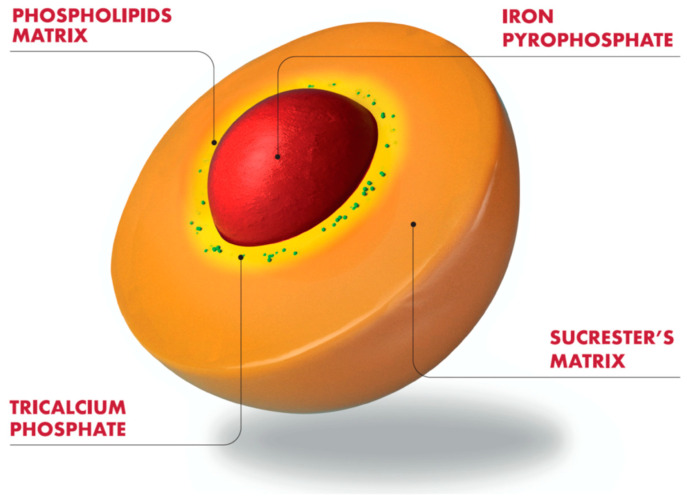
Schematic structure of Sucrosomial^®^ iron.

**Table 2 pharmaceuticals-16-00847-t002:** Laboratory assessment of iron status (adapted from reference [18]).

Clinical Setting	Laboratory Data	Diagnosis
Anemia and/or signs and symptoms suggestive of iron deficiency	Ferritin 30–300 ng/mL +TSAT > 20%	Iron repletion
	Ferritin < 30 ng/mL	Absolute iron deficiency
	Ferritin 30–100 ng/mL +TSAT < 20% or CRP > 5 mg/L	
	Ferritin >100 ng/mL +TSAT < 20% or CRP > 5 mg/L	Functional iron deficiency (iron sequestration) *
Blood donation/pregnancy	Ferritin < 50 ng/mL +TSAT > 20%	Inadequate iron stores
Major surgery (Blood loss > 500 mL)	Ferritin <100 ng/mL +TSAT > 20%	

CRP, C-reactive protein; TSAT, transferrin saturation. * See text for additional test requests.

**Table 3 pharmaceuticals-16-00847-t003:** Some studies on the use of Sucrosomial iron (SI) in different clinical settings (10 studies, 663 patients).

Author (Year) [Ref] Study Type	Patients	Treatment Compound (Dose) Duration	Baseline Hb (g/dL)	Final Hb (g/dL)	Baseline Ferritin (ng/mL)	Final Ferritin (ng/mL)	Baseline TSAT (%)	Final TSAT (%)	GI Side Effects
Parisi et al. (2017) [70] RCT	80 non-anemic Singleton pregnancy 12–14 weeks	Control (no iron) FS (30 mg/day) SI (14 mg/day) SI (28 mg/day Up to 6 weeks postpartum	12.0 11.9 12.0 11.9	11.6 11.8 12.0 12.6	46.6 43.7 52.4 52.6	31.3 43.1 40.8 49.8	27.6 26.7 28.1 26.5	25.6 26.7 29.5 28.8	0% 0% 0% 0%
Mafodda et al. (2017) [66] RCT pilot	64 patients with solid tumors	SI (30 mg/day) + DEPO 500 mcg/3 w FG (125 mg/wk) + DEPO 500 mcg/3 weeks 2 months	9.4 9.2	12.7 12.9	---	---	---	---	3% 0%
Pisani et al. (2014) [65] RCT	99 ND-CKD3-5	SI (30 mg/day) (*n* = 66) FG (125 mg/week IV, up to 1000 mg) (*n* = 33) 3 months	10.8 10.7	11.4 11.7	71 68	86 239	16.5 17.0	18.3 21.5	12% 18%
Bertani et al. (2021) [71] RCT	42 UC Mild-to-moderate anemia	SI (60 mg/day, 2 months, plus 30 mg/day, 1 month) FCM (1000 mg IV, at baseline) 3 months	11.1 10.3	12.2 11.8	16 10	26 131	---	---	5% 0%
Elli et al. (2018) [63] Observational	43 celiac disease	SI (30 mg/day) intolerant to FS (*n* = 24) FS (105 mg/day) (*n* = 17) 3 months	10.9 11.0	12.0 12.9	10.7 13.4	18.2 59.1	10.0 10.6	14.8 19.6	0% 10%
Farinati et al. (2018) [72] Observational	20 women with AIAG and anemia	SI (120 mg/daily, either fasting or during meals) 8 weeks	10.5	12.5	7	27	---	---	15%
Ciudín et al. (2017) [64] Case–control	40 bariatric surgeries All women	SI (28 mg/day) (*n* = 20) IS (300 mg IV) *(n* = 20) 3 months	12.4 12.5	12.3 12.7	102 98	89 96	22.9 23.6	24.1 26.3	0% 0%
Karavidas et al. (2021) [73] Observational	50 patients with HFrEF (LVEF 27 ± 5)	SI (28 mg/day), 3 months (*n* = 25) * Matched non-treated controls (*n* = 25) Follow-up 3 and 6 months	12.5 12.9	12.9 13.2 12.7 12.6	39 45	67 79 45 44	--- ---	--- ---	1 (4%) ---
Giordano et al. (2019) [74] RCT	135 patients with MSD and low-risk refractory anemia	SI (28 mg/day) + EPO (*n* = 54) FG (62.5 mg/week) + EPO (*n* = 43) No iron + EPO (*n* = 38) 3 months	8.9 9.3 9.7	12.0 12.0 12.0	610.2 608.8 699.5	607.0 ^#^ 723.7 ^#^ 730.3 ^#^	--- --- ---	32 ^#^ 48 ^#^ 40 ^#^	No
Giordano et al. (2021) [75] RCT	90 patients with IDA due to bleeding	SI (120 mg/day) (*n* = 45) FG (62.5 mg/day to cover TID) (*n* = 45) 4 weeks **	8.5 8.2	12.0 12.0	5 7	260 ^##^ 18 ^##^	--- ---	--- ---	36% 22% ^&^

AIAG, autoimmune atrophic gastritis; Hb, hemoglobin; RCT, randomized controlled trial; ND-CKD, non-dialysis-dependent chronic kidney disease; CHF, chronic heart failure; NYHA, New York Heart Association; LVEF, left ventricular ejection fraction; IDA, iron deficiency anemia; FS, ferrous sulfate; FG, ferric gluconate; DEPO, darbepoetin; TSAT, transferrin saturation; GI, gastrointestinal. * There were also improvements in quality of life, as assessed by the Kansas City Cardiomyopathy Questionnaire (from 55.7 to 61.8; *p* = 0.038), 6-min walking distance (from 318 m to 332 m; *p* = 0.065), and B-natriuretic peptide (from 643 to 535; *p* = 0.360). ** Treatment costs per month were 120 EUR/patient for SI and 300 EUR/patient for FG. ^#^ One month after enrolment; ^##^ 12 months after enrolment; ^&^ 5 urticaria and headache and 5 hypotension.

**Table 5 pharmaceuticals-16-00847-t005:** Incidence of gastrointestinal adverse effects in patients receiving oral Sucrosomial^®^ iron administration across clinical settings.

Clinical Setting	Patients Receiving Oral SI (*n*)	Gastrointestinal Adverse Effects (%) *
Obstetrics and Gynecology	165	5.4
Oncology	232	8.9
Gastroenterology	238	9.2
Nephrology	301	4.2
Cardiology	81	1.2
Internal Medicine	233	8.2
Surgery	762	1.2
*Overall*	*2012*	*4.5*

* Estimated from data in Table 3 and Table 4 and Appendix A.

## Data Availability

Data sharing not applicable.

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
