# Peer review of "Sucrosomial® Iron: An Updated Review of Its Clinical Efficacy for the Treatment of Iron Deficiency"

_pharmaceuticals, 2023, doi:10.3390/ph16060847_

Round 1

Reviewer 1 Report

I enjoyed reading the present review article by Gómez-Ramírez for its terse and exhaustive account of the topic. The figures and tables are also very clear. I have only a couple of comments/suggestions.

If the Authors deem them appropriate, I would suggest spending some paper on iron toxicity, in general, and another (separate) on the possible side-effects reported while using Sucrosomial (perhaps, in the form of an additional table). Just suggestions.

Author Response

Comment: I enjoyed reading the present review article by Gómez-Ramírez for its terse and exhaustive account of the topic. The figures and tables are also very clear. I have only a couple of comments/suggestions.

Answer: Thank you very much for your nice comment on our work.

Comment. If the Authors deem them appropriate, I would suggest spending some paper on iron toxicity, in general, and another (separate) on the possible side-effects reported while using Sucrosomial (perhaps, in the form of an additional table). Just suggestions.

Answer: Though it is an interesting suggestion, we do consider that “a more detailed analysis of the safety of the different options for iron replacement therapy is outside of the scope of this review, and can be found elsewhere [17,50]”.Nevertheless, we believe that an updated analysis of iron toxicity merits a full review on its own right.

We have also added a table with the estimated incidence of gastrointestinal adverse effects of oral sucrosomial iron, and the following sentence in the conclusion section “As depicted in Table 4, data from ≈2000 patients receiving oral SI supplementation, including a significant proportion of patients intolerant to FS, revealed a mean incidence of gastrointestinal adverse effects of 4.5%, which favorably compares with the much higher incidence reported for oral iron salts [33.34]”.

Reviewer 2 Report

Dear authors,

The review article is interesting and deals with Iron Deficiency in various clinical scenarios. The paper offers a good clinical overview of applications of. Oral (per os) or IV (intravenous) iron supplementation when the subject is suffering from ID or IDA.

I am not sure the article fits well the journal and would recommend another more appropriate journal like Medicina or Healthcare.

Indeed, the article does not report the different commercially available iron drugs (e.g. Ferrostrane/Ferrostrane commonly administred per os + Vitamin C, Ferrinject/Veninject used for IV perfusion) not their chemical structures pharmacological properties (bioavailability, dose-effect, toxicity etc) and mechanisms. 

the paper does not outlined the SI benefits and drawbacks/limitations in each clinical case. This should be implemented.

the paper does not relate the case of patients with sickle cell anemia/SCA (see below suggested references)

the review paper does not explain what is the benefit of SI when the patient is suffering from bad (mainly duodenal) absorption of iron … in other terms, does oral SI can be really benefit those patients, not only compared to iron salts (which I believe SI can be better option) but compared to IV iron ??

also, in clinical practice, does SI safe till which concentration/dose? At which frequency SI shall/can be used (according to the clinical case)?

some acronyms like IDA should be defined in the main text as first they appear 

Key referenceto be cited in relation to SCA and iron/iron overload (hematochromasis)

1- Menaa F. Stroke in sickle cell anemia patients: a need for multidisciplinary approaches. Atherosclerosis. 2013 Aug;229(2):496-503. doi: 10.1016/j.atherosclerosis.2013.05.006. 

I would consider the article in this journal after those revisions. 

best,

the reviewer 

I would consider the article in this journal after those major revisions, the paper being written mostly from the point of view of a clinician while it should be from a point of view pharmacological and pharmaceutical point of view, to fit the journal better… 

Author Response

Comment: The review article is interesting and deals with Iron Deficiency in various clinical scenarios. The paper offers a good clinical overview of applications of. Oral (per os) or IV (intravenous) iron supplementation when the subject is suffering from ID or IDA.

Answer: Thank you for your nice comments on our work

Comment: I am not sure the article fits well the journal and would recommend another more appropriate journal like Medicina or Healthcare.

Answer: This is an “invited review”  aimed at updating a previously published one in Pharmaceuticals in 2018 (ref 18), which was requested to us for an special issue of the Journal. Prior to submission, an outline of the present review was sent to the Editor-in-Chief of Pharmaceuticals, who agreed with the paper focus and content.

Comment: Indeed, the article does not report the different commercially available iron drugs (e.g. Ferrostrane/Ferrostrane commonly administred per os + Vitamin C, Ferrinject/Veninject used for IV perfusion) not their chemical structures pharmacological properties (bioavailability, dose-effect, toxicity etc) and mechanisms.

Amswer: This is not a general review of the different options of iron supplementation for treating ID/IDA, but an updated review of the clinical efficacy of Sucrosomial® iron for treating ID/IDA. The review was based on published studies where the efficacy of Sucrosomial® iron was tested in non-comparative studies (case series) or compared with other oral iron formulations (mostly ferrous sulfate, which has been long considered the gold standard for oral iron supplementation) and, in some studies, with IV iron formulations (ferric gluconate, iron sucrose, ferric carboxymaltose).

A more detailed description of structure and preclinical data of Sucrosomial® iron was presented in the previous review (ref 18), to which the reader is referred to.

Comment: The paper does not outlined the SI benefits and drawbacks/limitations in each clinical case. This should be implemented.

This has been summarized in the second paragraph of the conclusion section, which now reads as follows:

“Thus, the administration of oral SI emerges as a safe and efficacious first option for treating uncomplicated ID, especially for patients who are intolerant and/or refractory to traditional oral iron formulations. Moreover, oral SI should also be considered as an alternative option for initial and/or maintenance treatment of ID, in certain conditions usually treated by IV iron in current clinical practice, except in patients for whom immediate iron replenishment is required (e.g., IDA patients undergoing hip fracture repair surgery). However, as SI is commercialized as a nutritional supplement, there may be reimbursement issues in some countries”.

Comment: The paper does not relate the case of patients with sickle cell anemia/SCA (see below suggested references)

Answer: There are no data on the use of sucrosomial iron in sickle cell anemia, and therefore we have not commented on it.

Comment: The review paper does not explain what is the benefit of SI when the patient is suffering from bad (mainly duodenal) absorption of iron … in other terms, does oral SI can be really benefit those patients, not only compared to iron salts (which I believe SI can be better option) but compared to IV iron ??

At the begining of section 5, we stated that “….This structure allows SI to be gastro-resistant and to avoid the interaction between iron and intestinal mucosa, thus minimizing gastrointestinal side-effects (Figure 2). To date, in vitro studies have shown that SI absorption can occur through an hepcidin-independent pathway, as it is mostly taken up as vesicle-like nanoparticules, via paracellular and transcellular (M cells) routes which are not restricted to the duodenum and proximal jejunum [18]”

This preclinical data translated into clinical benefits, as described along the manuscript. For instance:

  • In gastroenterology, we have presented data on the efficacy of oral SI for treating IDA in patients who are intolerant and/or refractory to oral iron salts (e,g., patients with inflammatory bowel disease, celiac disease, atrophic autoinmune gastritis). In IBD patients, the efficacy of SI was compared to that of IV ferric carboxymaltose or iron sucrose.
  • The efficacy of SI has been shwon similar to that of IV iron in patients with cancer or myelodysplastic syndrome receiving ESA (vs. ferric gluconate), CKD (vs. ferric gluconate), severe bleeding-associated anemia (vs. ferric gluconate), postoperative anemia after cardiac (vs. ferric carboxymaltose) or bariatric surgery (vs. iron sucrose).
  • Oral iron therapy of little clinical value for treating ID in CHF, as demonstrated for oral iron polysaccharide (150 mg, bid) in the IRONOUT study (ref 107), and current guidelines recommend the IV route for iron supplementation (ref 105). However, compared to no iron, in 50 CHF patients with reduced ejection fraction (CHFrEF) and ID iron deficiency, with or without anemia, SI adminstration (28 mg/day, for 3 month) was associated with significant increase in hemoglobin, serum iron and serum ferritin concentrations along with a significant improvement in 6-min walked distance (6MWD) and Kansas City Cardiomyopathy Questionnaire (all p<0.01) (ref 108).

 Comment: Also, in clinical practice, does SI safe till which concentration/dose? At which frequency SI shall/can be used (according to the clinical case)?

The usual dose is 30 mg elemental iron once or twice in a day (1-2 capsules), depending on the severity of anemia (mild-to-moderate) and clinical context. The highest dose tested was 30 mg four times in a day (4 capsules) and it was given to patients with gastrointestinal or gynecologic bleeding-induced severe IDA (ref 114) or after cardiac surgery (ref 135), for whom SI was proved effective and well tolerated.

Comment: Some acronyms like IDA should be defined in the main text as first they appear.

Sorry for this mistake, that has been corrected.

Comments on the Quality of English Language

I would consider the article in this journal after those major revisions, the paper being written mostly from the point of view of a clinician while it should be from a point of view pharmacological and pharmaceutical point of view, to fit the journal better…

Anwer: As stated above, this is an “invited review”  aimed at updating a previously published one in Pharmaceuticals (2018), which was requested to us for an special issue of this Journal. Prior to submission, an outline of the present review was sent to the Editor-in-Chief of Pharmaceuticals, who agreed with the paper focus and content.